# Quality of life and its associated factors among epileptic patients attending public hospitals in North Wollo Zone, Northeast Ethiopia: A cross-sectional study

**Ayelign Mengesha Kassie**[1]*, **Biruk Beletew Abate**[1◉], **Mesfin Wudu Kassaw**[1◉], **Addisu Getie**[1‡], **Adam Wondmieneh**[1‡], **Kindie Mekuria Tegegne**[1‡], **Mohammed Ahmed**[2‡]

1 School of Nursing, College of Health Sciences, Woldia University, Woldia, Amhara, Ethiopia, 2 School of Public Health, College of Health Sciences, Woldia University, Woldia, Amhara, Ethiopia

◉ These authors contributed equally to this work.
‡ These authors also contributed equally to this work.
* Ayelignmengesha59@gmail.com

**Data Availability Statement:** All relevant data are within the paper and its Supporting Information files.

## Abstract

### Background

Epilepsy is thought to be caused by witchcraft, evil spirit, and God's punishment for sins in many developing countries. As a result, people with epilepsy and their families usually suffer from stigma, discrimination, depression, and other psychiatric problems. Thus, this study aimed to assess the quality of life and its associated factors among epileptic patients attending public hospitals in North Wollo Zone, Northeast Ethiopia.

### Methods

An institution-based cross-sectional study design was employed in this study. A simple random sampling technique was utilized. Health-related quality of life was measured based on the total score of the Quality of Life in Epilepsy Inventory (QOLIE-31) instrument. Data were entered into Epi-data 3.1 statistical package and exported to SPSS Version 20 for further analysis. Linear regression models were used to assess the relationship between quality of life and the independent variables. Statistically significant values were declared at a P-value of < 0.05.

### Results

A total of 395 patients participated in the study making the response rate 98.5%. The mean age of the participants was 32.39 ±10.71 years. More than half, 199 (50.4%) of epileptic patients had an overall weighted average health related quality of life score of mean and above. Male sex (B = 4.34, 95%CI, 0.41, 8.27, P = 0.03), higher educational status (B = 7.18, 95%CI, 1.39, 13.00, P = 0.015) and age at onset of epilepsy (B = 0.237, 95%CI, 0.02, 0.45, P = 0.035) were associated with increased health related quality of life score. On the other hand, family history of epilepsy (B = -4.78, 95%CI,-9.24,-0.33, P = 0.035), uncontrolled

**Funding:** The study was funded by Woldia University. Amount received =21,642 Ethiopian Birr.

**Competing interests:** The authors have declared that no competing interests exist.

**Abbreviations:** ASM, Anti-Seizure Medications; PWE, People Living With Epilepsy; SPSS, Software Package of Social Science; WHO, World Health Organization.

seizure (B $=$ -11.08, 95%CI,-15.11,-7.05, P < 0.001), more than 5 pre-treatment number of seizures (B $=$ -4.86, 95%CI,-8.91,-0.81, P $=$ 0.019), poor drug adherence (B $=$ -11.65, 95% CI,-16.06,-7.23, P < 0.001), having moderate (B $=$ -4.526, 95%CI,-8.59,-0.46, P $=$ 0.029) to sever (B = -12.84, 95%CI,-18.30,-7.37, P < 0.001) anxiety and depression, believing that epilepsy is caused by evil spirit (B $=$ -7.04, 95%CI,-11.46,-2.61, P $=$ 0.002), drinking alcohol (B $=$ -5.42, 95%CI,-10.72,-0.13, P = 0.045), and having other co-morbidities (B $=$ -9.35, 95% CI,-14.35,-4.36, P < 0.001) were significantly negatively associated with the health related quality of life score among epileptic patients.

## Conclusions

Only around half of the epileptic patients have a good health-related quality of life. In addition, multiple variables including family history, uncontrolled seizure, and poor drug adherence were associated with quality of life among epileptic patients. Hence, targeting these variables in epilepsy management is recommended.

## Background

Epilepsy is a chronic and non-communicable disease of the brain. It is characterized by recurrent seizure episodes involving part of the body or the entire body. Seizure episodes are due to the release of excessive electrical discharges from a group of brain cells [1]. The seizures can present in different forms ranging from a brief lapse of attention or muscle jerks to severe and prolonged convulsions. The episodes also vary in frequency and range from less than one per year to several seizures per day. Globally, more than 50 million people are estimated to have active epilepsy, and the annual cumulative incidence rate is estimated at 67.77 per 100,000 persons [2, 3].

The risk of premature death in people with epilepsy is also up to three times higher than in the general population. Many factors including stroke, brain tumor, head injury, central nervous system infections, and genetically inherited defects are believed to be the cause of epilepsy. Different types of anti-seizure medications (ASMs) are available and are highly effective for the treatment of Epilepsy. Approximately 70% of epileptic patients are estimated to successfully respond to proper treatment with ASMs [2–4].

Despite the fact that most epileptic patients can become free from seizure attacks with the optimal use of ASMS, the treatment outcome in the majority of cases remains insufficient and many suffer from uncontrolled seizure attacks especially in low-income countries [5]. The treatment outcome of epilepsy is associated with several factors. Gender of epileptic patients, age of the patient at seizure onset, type of epileptic seizure, seizure frequency, etiology of epilepsy, and presence of co-morbidities are among these factors [6–8].

Unlike developed countries, people in developing countries especially in Africa, don't have enough knowledge about epilepsy and is thought by many to be caused by witchcraft, evil spirit, and God's punishment for sins and they even do not seek medical care when they are sick. As a result, epileptic patients and their families usually suffer from stigma, discrimination, and prejudice [2, 9].

It is shown that patients with epilepsy and their families most of the time prefer traditional and religious healers and in house prayers [9]. Similarly, the self-management practice of epileptic patients is very low. Concerning this, studies revealed that in developing countries, the

rate of ASM therapy adherence is very poor [9, 10]. As a result, not only from recurrent seizure attacks but also patients with epilepsy are suffering from poor quality of life (QOL). QOL is a broad idea and includes the person's physical health, mental health, level of independence, social health, personal beliefs, and their relationship to each other and with the outside environment [11].

There is a tremendous negative impact caused by the poor adherence, low self-management practice, stigma, and other co-morbid conditions on patient's QOL, especially concerning the working conditions and personal and social relationships [12]. Therefore, psychosocial problems including anxiety, depression, social stigma, and deficiency of social backing, and unemployment can adversely affect their QOL [13–16].

Moreover, when patients took an ASM therapy for a certain time, that drug can provoke adverse effects and this can further impair their QOL [4, 17]. For these reasons, identifying predictors of reduced QOL in epileptic patients is critical to improving interventions and management strategies for epilepsy [18]. Such growing recognition of the importance of the psychosocial effects of epilepsy has led to the need to quantify QOL in affected individuals with epilepsy [16, 19].

Ethiopia shares the burden of epilepsy and over 1 million people are estimated to have active epilepsy [20]. The community's awareness of epilepsy in Ethiopia is not different from other African countries. As a result, epilepsy is believed to be caused by witchcraft, evil spirit, and God's punishment for sins by many [21, 22]. Consequently, patients with epilepsy in Ethiopia are suffering from physical disability, stigma, discrimination, and other psychiatric problems [10]. Despite this devastating impact, the public health community does not give epilepsy the attention it deserves. Hence, this study aimed to assess the quality of life and its associated factors among patients on anti-seizure medications therapy in North Wollo Zone public hospitals.

## Methods

### Study setting, design, and period

An institution-based cross-sectional was carried out in North Wollo zone public hospitals, Amhara region, Ethiopia. According to the 2007 Census conducted by the Central Statistical Agency of Ethiopia (CSA), North Wollo Zone has a total population of 1,500,303, of whom 752,895 are men and 747,408 women; with an area of 12,172.50 square kilometers [23]. The study participants were recruited from Kobo primary hospital, Lalibela primary hospital, and Woldia Comprehensive Specialized hospital. The study was conducted from February, 01 to June 30, 2020, among adult epileptic patients (age ≥ 18 years), who have been diagnosed with epilepsy, and are on at least one anti-seizure medication therapy in North Wollo zone public hospitals.

### Eligibility criteria

All adult epileptic patients (age ≥ 18 years), who have been on regular follow- up for at least one year with at least one anti-seizure medications therapy were included because seizure freedom 1 year after ASM treatment is suggested as a good predictor of long-term remission [24]. However, epileptic patients who had psychiatric problems, those who are seriously ill and unresponsive for the interview, and those who had hearing problems were excluded.

### Sample size determination

The sample size for this study was determined by using the single population proportion formula considering the assumptions: The proportion of epileptic patients being seizure-free,

38.6%, taken from a study in Addis Ababa (p = 0.386) because it provides the maximum sample size among the factors for quality of life [25]. Level of significance 5% (α = 0.05), Z α/2 = 1.96 and margin of error 5% (d = 0.05). The sample size was calculated as follows:

$$\mathbf{n_o} = \frac{\mathbf{Z\,(\alpha/2)^2 * p\,(1-p)}}{\mathbf{d^2}}$$

Hence, the minimum required sample size was = (1.96)2 * (0.386) (0.614)/(0.05)2 = **364.** Adding a 10% non-response rate, the final sample size appeared to be **401**.

## Sampling technique and procedure

A simple random sampling technique was employed to select the required number of epileptic patients attending public hospitals in the North Wollo zone. First, three representative hospitals were selected randomly. Then, in these selected hospitals, the number of study participants was assigned proportionally based on the number of epileptic patients in each of those hospitals. Finally, study participants were recruited consecutively by assuming patient flow as random.

## Operational definitions

**Quality of life** is defined as good if a participant scores mean and above the mean score of quality of life-related questions and, poor if the score is below the mean score of quality of life measuring questions.

   **Seizure control.**   The seizure status was considered to be controlled if the patient had not experienced any seizure attacks in the last one year and not-controlled if the patient experienced one or more seizure attacks in the last one-year follow-up period.

   **Anxiety and depression.**   The four-item patient health questionnaire for anxiety and depression (PHQ-4) was used to measure anxiety and depression among epileptic patients. The total score was determined by adding together the scores of each of the 4 items. Scores were rated as normal (0–2), mild (3–5), moderate (6–8), and severe (9–12) [26].

## Data collection technique and quality assurance

A structured and interviewer-administered Amharic version questionnaire was used to collect data from the study participants. The questionnaire was adapted from previous comparable studies [27, 28] and was first prepared in English. Then, it was translated to Amharic and back to English for consistency of the questions. The questionnaire has three sections. The first section was about the socio-demographic characteristics of the participants. Secondly, patients, medical records were assessed to extract additional information concerning the date of initiation of ASMs, types of epilepsy, and related issues clinical issues. The third section consists of the Quality Of Life in Epilepsy (QOLIE)-31questionner which is a survey question of health-related quality of life for adults (18 years or older) with epilepsy. Health-related quality of life was measured based on the total score of QOLIE-31questionner.

   A QOLIE- 31questionner consists of 31 items categorized under seven domains covering the following concepts of health: Overall Quality of Life (2 items), Medication Effects (3 items), Energy Fatigue (4 items), Seizure Worry (5 items), Emotional Well-being (5 items), Cognitive Function (6 items), and Social Function (5 items). The raw scores are rescaled from zero to 100 with higher values reflecting better QOL. Each item is scored on a scale of 0 to 100, with a score of zero representing the worst quality of life and a score of 100 representing a higher quality of life. However, possible response sets for scoring vary across questions. Examples of response sets used include (i) 0, 25, 50, 75, 100; (ii) 0, 20, 40, 60, 80, 100; and (iii) 0, 33,

67, 100 [29]. Each of the seven domains of QOL was scaled from 100%. Then, the overall weighted average quality of life score was calculated by summing the product of each of the seven domain scales times its weight and summing overall scales. Further details about the scoring of QOLIE-31questionnaire are available somewhere else [30].

Three BSc nurses were recruited for the data collection and three psychiatric nurses for supervision. A one-day training was given for the supervisors and data collectors on; the objectives of the study, the questions, and extent of explanations, the way to keep privacy and confidentiality, and other ethical issues. The supervisors coordinated the overall data collection process. A pre-test study was conducted on 5% of the sampled population by taking 23 epileptic patients at Dessie referral hospital which is not included in the study settings. The pre-test study was conducted 2 weeks before the actual data collection period. The study subjects who fulfill the inclusion criteria were selected by a simple random sampling technique. The result of the pretest was discussed and all the necessary amendments were made on the instructions, contents, order, and grammatical issues based on the pretest results. All data were checked for completeness, accuracy, clarity, and consistency by the supervisors and the investigators each night after the data were collected. Investigators and supervisors closely monitored the data collection process. Double data entry and validation were performed and data were intensively cleaned before analysis.

## Data processing and analysis

Data were coded and entered into the Epi-data version 3.1statistical program and then exported to SPSS Version 20 for further analysis. Descriptive statistical methods were computed for the study variables. A simple linear regression model was used to assess the relationship between quality of life and the independent variables and then, a multiple regression analysis was run to assess the association between independent variables while controlling confounders. Linear regression model assumptions including variable measurement at the continuous level, the presence of a linear relationship between the independent and the outcome variables, absence of significant outliers, independence of observations, homogeneity of variance, normally of distribution among the errors, and absence of multicollinearity among the independent variables were assessed and all of them were fulfilled. Finally, a P-value of less than 0.05 was taken as a cut-off point to declare statistically significant associations between the independent variables and the quality of life score among the epileptic patients.

## Ethical approval and consent to participant

Ethical clearance was obtained from Woldia University institutional review board with an approval number of wldu/012/2020 and support letters were written to the selected public hospitals and concerned others to obtain permission and cooperation during the data collection process. Written informed consent was obtained from each study participant before the data collection process. Confidentiality of the information was preserved and the privacy of the respondents was maintained by making the questionnaires anonymous and putting them in secured places after the data were collected.

## Results

### Sociodemographic and behavioral characteristics of respondents

A total of three hundred ninety-five participants volunteered and provided a complete response for the interview making the response rate 98.5%. The mean age of the participants was 32.39 with a standard deviation of 10.71 years. More than half, (58.5%) of them were females. The majority, that is 255 (64.6%) of the participants were orthodox Christianity

followers and 130 (32.9%) were Muslims. Regarding marital status, 187 (47.3%) were married, 146 (37.0%) single, 49 (12.4%) divorced and the remaining 13 (3.3%) were widowed. More than half, 223 (56.5%) of the participants were rural dwellers.

Regarding education, 203 (51.4%) of the participants have no formal education, and the remaining 49.6% have completed up to higher levels of education. Occupationally, 133 (33.7%) of the participants were farmers, 52 (13.2) government employee, 40 (10.1%) merchant, 79 (20.0%) student, and the remaining, 91(23.0%) fall in other categories (housewife, daily laborer, unemployed) (Table 1).

## Clinical characteristics of study participants

Eighty-four (21.3%) of the study participants had a family history of epilepsy. Around 60 percent of the participants had a generalized type of epileptic seizure and the remaining had a focal type of seizure. Thirty-seven percent of the participants have claimed that they have lived with the disease for more than one year before seeking medical treatment. About 73.0% of the patients have attended epilepsy treatment follow up for more than 2 years. Regarding treatment regimen, around 59.0% of the patients were on polytherapy for epilepsy. In addition, only 44.8% of the patients were found to be adherent to ASMs. All patients were found to have anxiety and depression ranging from mild to severe levels. Furthermore, 55.2% of the epileptic patients had uncontrolled seizure status (Table 2).

## Health-related quality of life among epileptic patients

The average total quality of life score of the seven domains of health-related quality of life among epileptic patients was as follows: Seizure worry (63.43±26.56); Overall quality of life

**Table 1. Sociodemographic and behavioral characteristics of epileptic patients ($n$ = 395).**

| Characteristics | Category | Frequency (%) |
|---|---|---|
| Age, years | Mean±SD (32.39±10.71) | |
| Sex | Male | 231(58.5) |
| Religion | Orthodox | 255(64.6) |
| | Muslim | 130(32.9) |
| | Protestant | 10(2.5) |
| Marital status of participants | Married | 187(47.3) |
| | Single | 146(37.0) |
| | Divorced | 49(12.4) |
| | Widowed | 13(3.3) |
| Residence | Urban | 172(43.5) |
| Educational status of Participants | No education | 203(51.4) |
| | Primary | 74(18.7) |
| | Secondary | 66(16.7) |
| | Higher | 52(13.2) |
| Occupation of Participants | Gov't employee | 52(13.2) |
| | Student | 79(20.0) |
| | Farmer | 133(33.7) |
| | Merchant | 40(10.1) |
| | Others | 91(23.0) |
| Alcohol drinking habit | Yes | 82(20.8) |
| Smoke cigarette | Yes | 23(5.8) |
| Chew chat | Yes | 45(11.4) |

**Table 2. Clinical characteristics of epileptic patients (*n* = 395).**

| Characteristics | Category | Frequency (%) |
|---|---|---|
| Family history of epilepsy | Yes | 84(21.3) |
| Type of epileptic seizure | Generalized | 236(59.7) |
| Age of the patient at the onset of epilepsy | <30 years | 300(75.9) |
| | 30-45years | 78(19.7) |
| | >45 years | 17(4.3) |
| Pre-treatment duration with epilepsy | ≤ 12 month | 249(63.0) |
| Pre-treatment number of seizures | ≤ 5 seizures | 125(31.6) |
| Duration of epilepsy treatment follow up | < = 2 years | 107(27.1) |
| Epilepsy treatment regimen | Monotherapy | 163(41.3) |
| Believe that epilepsy is caused by an evil spirit | Yes | 93(23.5) |
| Adherence to ASMs | Good | 177(44.8) |
| Level of anxiety and | Mild | 169(42.8) |
| Depression | Moderate | 152(38.5) |
| | Severe | 74(18.7) |
| Other co-morbidity | Yes | 73(18.5) |
| Seizure control status | Uncontrolled | 218(55.2) |

(58.48±21.22); Emotional wellbeing (62.73±18.34), Energy/fatigue (58.00±20.48), Cognitive function (73.03±23.01), Medication effects (71.49±25.62), and Social functioning (74.97 ±22.56). The overall weighted average health-related quality of life score was 86.54±24.31. More than half, 199 (50.4%) of epileptic patients have an overall weighted average quality of life score above or equal to the mean score level, and the remaining,196 (49.6%) had an overall weighted average quality of life score below the mean.

## Factors associated with health-related quality of life

In the bivariate linear regression analysis; participants age, educational status, family history of epilepsy, epilepsy treatment follow-up duration, having an uncontrolled seizure, pre-treatment number of seizure, poor adherence to anti-seizure medications, having a severe level of anxiety and depression, poor sleep pattern, believing that epilepsy is caused by an evil spirit, drinking alcohol, cigarette smoking, chewing chat and having other co-morbidities were significantly associated with the health-related quality of life among epileptic patients.

On the multivariable linear regression analysis model, about 50.8% of the total variation in health related quality of life among epileptic patients was explained by the variables included in the model. Male sex (B $_=$ 4.34, 95%CI: 0.41, 8.27, P $_=$ 0.03), higher educational status (B $_=$ 7.18, 95%CI: 1.39, 13.00, P $_=$ 0.015), and age at onset of epilepsy (B $_=$ 0.237, 95%CI: 0.017, 0.45, P $_=$ 0.035), were associated with increased health related quality of life score among epileptic patients.

On the other hand, having family history of epilepsy (B $_=$ -4.78,95%CI: -9.24,-0.33, P = 0.035), having uncontrolled seizure (B $_=$ -11.08, 95% CI:-15.11,-7.05, P < 0.001), pre-treatment number of seizures (B $_=$ -4.86, 95%CI:-8.91,-0.81, P $_=$ 0.019), poor adherence to anti-seizure medications (B $_=$ -11.65, 95%CI: -16.06,-7.23, P < 0.001), having moderate anxiety and depression (B $_=$ -4.526, 95%CI: -8.59, -0.46, P $_=$ 0.029), having sever anxiety and depression (B $_=$ -12.84, 95CI: -18.30,-7.37, P < 0.001), believing that epilepsy is caused by evil spirit (B $_=$ -7.04, 95%CI: -11.46,-2.61, P $_=$ 0.002), drinking alcohol (B $_=$ -5.42,95%CI:-10.72,-0.13, P $_=$ 0.045), and having other co-morbidities (B $_=$ -9.35, 95%CI: -14.35,-4.36, P < 0.001) were

significantly negatively associated with the health related quality of life among epileptic patients (Table 3).

## Discussion

In this study, around half, 49.6% of epileptic patients had poor quality of life. This finding is consistent with study findings from Jimma University (49.7%) and Addis Ababa at Amanuel specialized hospital (45.8%) [27, 28]. It is also in line with a study conducted at the University of Gondar Referral Hospital which has revealed that 44.2% of participants were found to have a poor quality of life [31]. Researchers in Italy have reported a consistent finding that 43.3% of patients had poor quality of life at a health institution study [32].

However, it is higher than a study at Ambo general hospital that 24.4% had an overall poor health-related quality of life [33]. This variation might be due to differences in the tool and operational definitions. Unlike, the current study, the previous study has used the World Health Organization Quality of Life (WHOQOL) tool for measuring the health-related quality of life among epileptic patients which is not specific for epilepsy while, the current study was conducted using the Quality of Life Inventory for Epilepsy (QOLIE-31) a specifically designed tool to measure health-related quality of life among epileptic patients [30].

Regarding the quality of life score of the seven domains of QOLIE-31, participants scored the lowest average score on the Energy/fatigue domain, and the highest score was found in the Social functioning domain. A similar study has reported a consistent finding that participants have scored the highest mean under the social functioning domain [27]. This means patients with epilepsy are having an acceptable social interaction with the community which is encouraging. However, the low score under the energy/fatigue domain is worrying because it is an indication that patients with epilepsy are suffering from tiredness, loss of energy, and related impacts on their life [33].

Regarding the factors related to the quality of life, male sex, attaining higher education, and ages at the onset of epilepsy were positively associated with quality of life among epileptic patients. Being male was associated with increased quality of life scores among epileptic patients. This finding is in line with a study in Portugal that the female gender was associated with poorer health-related in patients with refractory epilepsy [34]. A consistent finding was also reported in India that the female gender was associated with reduced quality of life among epileptic patients [35]. This might be due to the effect of hormonal changes occurring throughout a woman's life which can influence and be influenced by seizure mechanisms and ASMs, presenting unique management challenges and this, in turn, may affect their quality of life [36]. It might be also due to a lack of social support. Unlike men, women in Ethiopia usually stay at home and work in the house and might not get sufficient social support from the community.

Having higher educational status was also associated with increased quality of life among epileptic patients. This finding is supported by studies conducted at Ambo general hospital [33], Mekelle City, Northern Ethiopia [37], Uganda [38], Kenya [39], and Iraq [40]. A consistent finding was also obtained in Indonesia that level of education was one of the positive predictors of quality of life in patients with epilepsy [41]. This could be due to the influence of education on individual perception of their disease condition and adherence to their medications. It might be also due to employment opportunities because people with higher educational status are highly likely to have different job offers and be employed by both governmental and non-governmental organizations.

As the age of onset of epilepsy increases, the quality of life of patients was also increased. This is supported by a study conducted in Malaysia [42]. A person with epilepsy diagnosed in

**Table 3. Linear regression analysis of factors for health-related quality of life in epileptic patients (*n* = 395).**

| Independent Variables | Crude Beta | 95% CI for Beta | P-value | Adjusted Beta | 95% CI for Beta | P-value |
|---|---|---|---|---|---|---|
| Age | | | | | | |
| Mean±SD (32.39±10.71) | -0.29 | (-0.51,-0.07) | 0.011 | -0.25 | (-0.53,0.02) | 0.072 |
| Sex | | | | | | |
| Male | 3.55 | (-1.33,8.42) | 0.153 | **4.34** | **(0.41,8.27)** | **0.030** |
| Female | 1 | | | | | |
| Educational status | | | | | | |
| No education | 1 | | | | | |
| Primary | 1.02 | (-5.15,7.19) | 0.75 | 2.91 | (-2.23,8.06) | 0.267 |
| Secondary | 2.75 | (-3.70,9.19) | 0.403 | 3.05 | (-2.60,8.69) | 0.289 |
| Higher | 5.41 | (-1.70,12.51) | 0.135 | **7.18** | **(1.39,13.00)** | **0.015** |
| Marital status | | | | | | |
| Married | 7.16 | (2.39,11.93) | 0.003 | 7.21 | (-2.99,17.42) | 0.165 |
| Single | 1.21 | (-3.78,6.19) | 0.635 | 0.81 | (-9.97,11.60) | 0.882 |
| Divorced | -14.83 | (-21.99–7.68) | <0.001 | -3.19 | (-14.34,7.96) | 0.574 |
| Widowed | 1 | | | | | |
| Family history of epilepsy | | | | | | |
| Yes | -9.44 | (-15.25,-3.64) | 0.002 | **-4.78** | **(-9.24,-0.33)** | **0.035** |
| No | 1 | | | | | |
| Age at onset of epilepsy in years (Mean±SD, 24.62±12.58) | 0.15 | (-0.05,0.34) | 0.138 | **0.24** | **(0.02,0.46)** | **0.035** |
| Treatment follow-up in years (Mean±SD, 5.20±4.20) | 0-.99 | (-1.56,-.43) | .001 | -0.12 | (-0.60,0.36) | 0.624 |
| Uncontrolled seizure | -20.49 | (-24.89.-16.10) | <0.001 | **-11.08** | **(-15.11–7.05)** | **<0.001** |
| Epilepsy type | | | | | | |
| Generalized | 4.03 | (-0.86,8.93) | 0.106 | -1.62 | (-5.39,2.14) | 0.397 |
| Focal | 1 | | | | | |
| Pre-treatment number of seizures | | | | | | |
| > = 5times | -7.270 | (-12.40,-2.14) | 0.006 | **-4.86** | **(-8.91,-0.81)** | **0.019** |
| < 5times | 1 | | | | | |
| Treatment regimen | | | | | | |
| Monotherapy | 1 | | | | | |
| Polytherapy | 3.64 | (-1.242,8.51) | 0.144 | -0.72 | (-4.44,3.00) | 0.703 |
| Shortage of ASMs | 0.32 | (-4.68,5.32) | 0.900 | -2.21 | (-6.07,1.65) | 0.260 |
| Adherence to ASMs | | | | | | |
| Good | 1 | | | | | |
| Poor | -25.86 | (-29.97,-21.76) | <0.001 | **-11.65** | **(-16.06,-7.23)** | **<0.001** |
| Anxiety and depression | | | | | | |
| Mild | 1 | | | | | |
| Moderate | 0.66 | (-4.28,5.61) | 0.792 | **-4.53** | **(-8.59,-0.46)** | **0.029** |
| Severe | -25.35 | (-30.99,-19.72) | <0.001 | **-12.84** | **(-18.30,-7.37)** | **<0.001** |
| Sleep pattern | | | | | | |
| Poor | 9.66 | (4.78,14.55) | <0.001 | 2.97 | (-0.10,6.94) | 0.142 |
| Good | 1 | | | | | |
| Belief that epilepsy is due to an evil spirit | | | | | | |
| Yes | -16.80 | (-22.23,-11.38) | <0.001 | **-7.04** | **(-11.46,-2.61)** | **0.002** |
| No | 1 | | | | | |
| Alcohol drinking | | | | | | |
| No | 1 | | | | | |
| Yes | -14.50 | (-20.26,-8.74) | <0.001 | **-5.42** | **(-10.72,-0.13)** | **0.045** |

*(Continued)*

Add

**Table 3.** (Continued)

| Independent Variables | Crude Beta | 95% CI for Beta | P-value | Adjusted Beta | 95% CI for Beta | P-value |
|---|---|---|---|---|---|---|
| Cigarette smoking | | | | | | |
| No | 1 | | | | | |
| Yes | -10.73 | -20.96,-0.50) | 0.040 | 5.13 | (-4.37,14.63) | 0.289 |
| Chat chewing | | | | | | |
| No | | | | | | |
| Yes | -7.96 | (-15.50,-0.42) | 0.039 | -3.10 | (-10.60,4.40) | 0.417 |
| Co-morbidity | | | | | | |
| No | | | | | | |
| Yes | -16.31 | (-22.30,-10.32) | <0.001 | **-9.35** | **(-14.35–4.36)** | **<0.001** |

Model fitness: $R^2$ = 50.8%

childhood would likely have many schooling hours disrupted if the frequency of seizures were uncontrolled and this could also lead to low educational achievement, and thus a lower employability potential. Furthermore, the seizure itself may cause accidents and injuries thus preventing sufferers from driving and depriving many types of employment opportunities, this, in turn, has the potential of negatively affecting the quality of life of epileptic patients [43].

On the other hand, the present study revealed that having a family history of epilepsy, having an uncontrolled seizure, pre-treatment number of seizures, poor adherence to ASMs, believing that epilepsy is caused by an evil spirit, drinking alcohol, having moderate to severe anxiety and depression and having other co-morbidities were negative predictors of quality of life among epileptic patients.

Family history of epilepsy was associated with decreased quality of life among epileptic patients. This is supported by a study conducted among children and adolescents in Iran which has reported that a family history of epilepsy was significantly associated with health-related quality of life [44]. A consistent finding was also found in a study conducted among patients with refractory mesial temporal lobe epilepsy in Brazil [45]. This might be due to the influence of perceived stigma among epileptic patients. Because, when there is epilepsy in a family, the community might have stigmatizing behavior towards the family as epilepsy is believed to be caused by God's curse, evil spirit, and witchcraft by many [46]. Furthermore, people might see epilepsy as a contagious disease, which in turn may result in a lack of social support, stigma, and discrimination among patients [47, 48].

The present study showed that pre-treatment number of seizures more than five times, and having uncontrolled seizures have negatively affected the quality of life of epileptic patients. The finding is supported by studies conducted in Ethiopia [37, 49], Uganda [38], and Iraq [40]. The possible reason could be because of seizure frequency, a known risk factor that can cause excessive fear, inability to work, stigmatization, diminished hope, and future life, impairment in social functions, and psychological impairments [42]. Individuals with frequent epileptic seizures will always be in discomfort because; they do not know when the next seizure might happen. As a result, they might take extra-care and impose self-restrictions from driving, cooking, and doing risky jobs to avoid having seizure episodes at inappropriate times, places, or social events [50, 51]. This in turn has a tremendous potential of compromising the quality of life of epileptic patients.

Having moderate to severe anxiety and depression, and other co-morbidities resulted in decreased quality of life in epileptic patients. This finding is supported by studies conducted in China [52], Ethiopia [31, 49], and Kenya [53]. This could be due to the effect of these problems

on quality of life, particularly, in the domains of lethargy/fatigue, and emotional well-being [53]. It might be also due to pill burden and drug interactions resulting from concomitant treatments [54]. Believing that epilepsy is caused by evil spirits negatively affects the quality of life of epileptic patients. This could be due to the influence of cultural beliefs on attitudes and uncaring actions of the community towards patients with epilepsy [55] which in turn can result in high rates of low self-esteem in patients with epilepsy, thus adversely affecting the quality of life [56].

Furthermore, drinking alcohol was associated with a reduced quality of life among epileptic patients. This might be due to the effect of alcohol on the frequency of seizure attacks resulting in a possible negative effect on the quality of life. Because, alcohol is a known precipitating factor for seizure disorders [57, 58]. Besides, alcohol consumers might have alcohol-drug interactions and this might further result in recurrent seizure attacks among patients [59, 60]. This might be also due to poor adherence to medications as a result of prolonged treatment for epilepsy with ASMs. Poor drug adherence is a known factor that was found to have a negative association with health quality of life among epileptic patients in this and many other studies, and, might result in a poor quality of life if patients are non-adherent to ASMs.

## Conclusion

In this study, only around half of the epileptic patients had a good quality of life. Furthermore, multiple variables including family history, uncontrolled seizure, poor adherence, and having anxiety and depression were associated with health-related quality of life among epileptic patients. Hence, targeting these variables in epilepsy management is essential. Moreover, early recognition of adherence issues and detecting and managing co-morbid conditions by the clinician can have a great impact on controlling seizure attacks and increasing the quality of life of epileptic patients.

## Supporting information

**S1 Data. SPSS data file.**
(SAV)

## Acknowledgments

The authors would like to express our gratitude to the data collectors and supervisors for their great contribution. Our deepest gratitude also goes to those who participated in this study.

## Author Contributions

**Conceptualization:** Ayelign Mengesha Kassie, Biruk Beletew Abate, Mesfin Wudu Kassaw, Addisu Getie, Adam Wondmieneh, Kindie Mekuria Tegegne, Mohammed Ahmed.

**Data curation:** Ayelign Mengesha Kassie.

**Formal analysis:** Ayelign Mengesha Kassie, Biruk Beletew Abate, Mesfin Wudu Kassaw, Addisu Getie, Adam Wondmieneh, Kindie Mekuria Tegegne, Mohammed Ahmed.

**Funding acquisition:** Ayelign Mengesha Kassie, Biruk Beletew Abate.

**Investigation:** Ayelign Mengesha Kassie.

**Methodology:** Ayelign Mengesha Kassie, Biruk Beletew Abate, Mesfin Wudu Kassaw, Addisu Getie, Adam Wondmieneh, Kindie Mekuria Tegegne, Mohammed Ahmed.

**Project administration:** Ayelign Mengesha Kassie.

**Resources:** Ayelign Mengesha Kassie.

**Supervision:** Ayelign Mengesha Kassie.

**Validation:** Ayelign Mengesha Kassie, Biruk Beletew Abate, Mesfin Wudu Kassaw, Addisu Getie, Adam Wondmieneh, Kindie Mekuria Tegegne, Mohammed Ahmed.

**Visualization:** Ayelign Mengesha Kassie.

**Writing – original draft:** Ayelign Mengesha Kassie, Biruk Beletew Abate, Mesfin Wudu Kassaw, Addisu Getie.

**Writing – review & editing:** Ayelign Mengesha Kassie, Biruk Beletew Abate, Mesfin Wudu Kassaw, Addisu Getie, Adam Wondmieneh, Kindie Mekuria Tegegne, Mohammed Ahmed.

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
