## [Decision Letter · Decision Letter 0]

17 Dec 2020

PONE-D-20-34326

Quality of life and its associated factors among epileptic patients attending public hospitals in north Wollo Zone, northeast Ethiopia: Across-sectional study

PLOS ONE

Dear Dr. Kassie,

Thank you for submitting your manuscript to PLOS ONE. After careful consideration, we feel that it has merit but does not fully meet PLOS ONE’s publication criteria as it currently stands. Therefore, we invite you to submit a revised version of the manuscript that addresses the points raised during the review process.

Specifically, you should clarify data collection and analysis in several points as indicated by the Reviewers.

We look forward to receiving your revised manuscript.

Kind regards,

Emilio Russo

Academic Editor

PLOS ONE

2. Please include in your Methods section (or in Supplementary Information files) the participating hospitals/institutions.

4. Please provide further clarification whether IRB approval was obtained from all participating hospitals prior to data collection.

5. Please ensure that you include a title page within your main document. We do appreciate that you have a title page document uploaded as a separate file, however, as per our author guidelines (http://journals.plos.org/plosone/s/submission-guidelines#loc-title-page) we do require this to be part of the manuscript file itself and not uploaded separately.

6. Thank you for stating the following in the Funding Section of your manuscript:

"The study was funded by Woldia University"

"Data supporting the conclusions of this article are included within the article and its supporting files. "

7. Your ethics statement should only appear in the Methods section of your manuscript. If your ethics statement is written in any section besides the Methods, please move it to the Methods section and delete it from any other section. Please ensure that your ethics statement is included in your manuscript, as the ethics statement entered into the online submission form will not be published alongside your manuscript.

Reviewers' comments:

Reviewer's Responses to Questions

**Comments to the Author**

1. Is the manuscript technically sound, and do the data support the conclusions?

Reviewer #1: Partly

Reviewer #2: Partly

2. Has the statistical analysis been performed appropriately and rigorously? 

Reviewer #1: Yes

Reviewer #2: Yes

3. Have the authors made all data underlying the findings in their manuscript fully available?

Reviewer #1: No

Reviewer #2: No

4. Is the manuscript presented in an intelligible fashion and written in standard English?

Reviewer #1: Yes

Reviewer #2: No

5. Review Comments to the Author

Reviewer #1: Based on my review results the manuscript is partly technically sound, and do the data support the conclusions because few informations on result and conclusion were has discrpancy with the data put on the table. The authers used appropriat model for thire problem however I have not get any information how they check the model fitness. Regards to data availablity the authors have not made all data underlying the findings in their manuscript fully available. eg SPSS data. Some edditorial issues are avaialable related to English language.

Reviewer #2: The manuscript describes quality of life and its associated factors among epileptic patients attending public hospitals in North Wollo zone, Ethiopia. There are several major points that need to be addressed and revised by the authors first before accepting the manuscript for publication. Moreover, line spacing and line numbering were not applied as per the PLoS one guideline. Hence it is difficult to evaluate the manuscript point by point. The manuscript has several grammar and punctuation errors that need extensive English language revision.

Title

Make space “a/cross-sectional”

Abstract

The conclusion you made is just a summary of the result. Need revision

Introduction

Several studies have been conducted in Ethiopia including in Jima University, in Gondar, Ambo….What is new in your study? and please clearly state in the introduction section.

Method

Why at least one AED use and 1 year since diagnosis were considered as eligibility criteria?

Hoe simple random sampling was carried? How do you access the sampling frame?

You have stated “…questionnaire was adapted from previous comparable studies..” please cite some of them.

Who collects the actual data (their qualification…)?

Result section

“…three hundred ninety five (395)…” no need to repeat the same information

” ….Christianity followers followed by 130 (32.9%).” It is an incomplete statement and revise it.

How did you collet seizure type? (did you use medical chart review)?

The proportion of medication adherence is not included in the result section

6. PLOS authors have the option to publish the peer review history of their article (what does this mean?). If published, this will include your full peer review and any attached files.

Reviewer #1: **Yes: **Agumas Fentahun Ayalew, Master Of Public Health in Epidemiology, Salale University, Health Science College, Department of Public Health, Fitche, Ethiopia

Reviewer #2: No

---

## [Author Response · Author response to Decision Letter 0]

23 Dec 2020

To: Plose One Editorial Team

Subject: Submitting a revised research manuscript for publication 

Dear Sir /Madam 

Greetings,

 I am Ayelign Mengesha from Woldia University, Ethiopia. I hereby submit a revised manuscript entitled “Quality of life and its associated factors among epileptic patients attending public hospitals in North Wollo Zone, Northeast Ethiopia: a cross-sectional study” to be published at Plose One. 

 First of all I would like to express my deepest gratitude on behalf of the authors for the very detail, genuine and constructive comments raised by the reviewer/s and, or the editor which I believe has improved the manuscript tremendously. We believe that we have addressed the points raised in the review process properly. The modifications and, or corrections made during the revision are clarified point by point below. 

 Yours sincerely,

 On behalf of the authors

General corrections/modifications:

Table has been included for clinical variables. 

Grammatical, spelling and punctuation mark issues are corrected with grammerly proof reading tool and with intense revision by the authors.

Other corrections/modifications are described in detail under point by point responses below.

Editor comments

Comment: Please ensure that your manuscript meets PLOS ONE's style requirements, including those for file naming. 

Author response: The manuscript has been revised intensively in line with the journal’s requirements.

Comment: Please include in your Methods section (or in Supplementary Information files) the participating hospitals/institutions.

Author response: Revised in methods section paragraph 1 as follows: The study participants were recruited from Kobo primary hospital, Lalibela primary hospital, and from Woldia Comprehensive Specialized hospital. The study was conducted from February, 01 to June 30, 2020. 

Comment: We suggest you thoroughly copyedit your manuscript for language usage, spelling, and grammar. 

Author response: Grammatical, spelling and punctuation mark issues are corrected with grammerly online proofreading and with intense revision by the authors. 

Comment: Please provide further clarification whether IRB approval was obtained from all participating hospitals prior to data collection. 

Author response: Ethical approval and consent to participant: Ethical clearance was obtained from Woldia University institutional review board and support letters were written to the selected public hospitals and concerned others to obtain permission and cooperation during the data collection process. Methods section, last paragraph. However, ethical approval from hospitals was not applicable because the hospitals but, permission to conduct the study in the hospitals was obtained.

Comment: Please ensure that you include a title page within your main document. We do appreciate that you have a title page document uploaded as a separate file, however, as per our author guidelines (http://journals.plos.org/plosone/s/submission-guidelines#loc-title-page) we do require this to be part of the manuscript file itself and not uploaded separately. Could you therefore please include the title page into the beginning of your manuscript file itself, listing all authors and affiliations?

Author response: I apologize for failing to include the title page in the main document. I have included the title page with author details within the main document. 

Comment: Thank you for stating the following in the Funding Section of your manuscript: "The study was funded by Woldia University". We note that you have provided funding information that is not currently declared in your Funding Statement. However, funding information should not appear in the Acknowledgments section or other areas of your manuscript. We will only publish funding information present in the Funding Statement section of the online submission form. Please remove any funding-related text from the manuscript and let us know how you would like to update your Funding Statement. Currently, your Funding Statement reads as follows: "Data supporting the conclusions of this article are included within the article and its supporting files". Please include your amended statements within your cover letter; we will change the online submission form on your behalf.

Author response: Funding should be written in the following ways: The study was funded by Woldia University. Amount received =21,642 Ethiopian Birr. 

Comment: The manuscript describes quality of life and its associated factors among epileptic patients attending public hospitals in North Wollo zone, Ethiopia. There are several major points that need to be addressed and revised by the authors first before accepting the manuscript for publication. Moreover, line spacing and line numbering were not applied as per the PLoS one guideline. Hence it is difficult to evaluate the manuscript point by point. The manuscript has several grammar and punctuation errors that need extensive English language revision. 

Author response: Dear Agimas Fentahun Ayalew, I thank you very much for your genuine and constructive comments: I have revised the manuscript intensively. Grammatical and punctuation mark issues corrected with grammerly proof editing and with intense revision by the authors.

Title 

Comment: Make space “a/cross-sectional” 

Author response: Space added, a cross-sectional study

Abstract 

Comment: The conclusion you made is just a summary of the result. Need revision

Author response: It has been revised with adjusted beta and confidence interval in the result section and conclusion. Page 2.

Introduction 

Comment: Several studies have been conducted in Ethiopia including in Jima University, in Gondar, Ambo….What is new in your study? and please clearly state in the introduction section.

Author response: Thank you for your view, as you might have seen in the result section, several variables were included in this study. In addition, many of the studies have used WHO quality of life tool to assess quality of life among epileptic patients which is general and is not specific in terms of content. But, in this study, we have used the Quality of life in epilepsy inventory; QOLIE-31 questionnaire which is designed to measure quality of life among epileptic patients. The study area is also quite different compared with other studies. It is not sufficiently addressed with health care services.

Method

Comment: Why at least one AED use and 1 year since diagnosis were considered as eligibility criteria?

Author response: The study participants were included if they were on treatment with AEDs because adherence was one of the variables which was included in this study. But, if a patient who has stopped taking drugs because he has been seizure free and was advised to stop taking the drugs by physicians comes to the hospitals for consultation and counseling, he will not be included. At least one AED was mentioned here because it is recommended to start treatment with one AED, and if patients fail to respond for that drug, then they will take additional drugs accordingly. 

At least one year was treatment was also another criterion to be included because seizure freedom 1 year after AED treatment is suggested as a good predictor of long-term remission (reference 24). If patients gain long term remission (being seizure free 2-3 years), then it is possible to discontinue the drugs and follow the patient for possible seizure relapse. Some other studies have used the same criteria. Page 5.

Comment: How simple random sampling technique was carried? How do you access the sampling frame?

Author response: Revised, patient flow was assumed as random and it is written as follows: finally, study participants were recruited consecutively by assuming patient flow as random. Page 5 last paragraph.

Comment: You have stated “…questionnaire was adapted from previous comparable studies.” please cite some of them.

Author response: Citations added. It is revised as follows: The questionnaire was adapted from previous comparable studies (27, 28). The questionnaire was first prepared in English. Then, it was translated to Amharic and back to English for consistency of the questions. Page 6.

Comment: Who collects the actual data (their qualification…)? 

Author response: It revised as follows: To ensure the quality of data, three BSc nurses were recruited for the data collection. In addition, three psychiatric nurses were also recruited for supervision. Page 7.

Result section 

Comment: “…three hundred ninety five (395)…” no need to repeat the same information

Author response: Removed

Comment:” ….Christianity followers followed by 130 (32.9%).” It is an incomplete statement and revise it.

Author response: Thank you; it is revised as: Majority, that is 255 (64.6%) of the participants were orthodox Christianity followers and 130 (32.9%) were Muslims. Page 8.

Comment: How did you collet seizure type? (did you use medical chart review)?

Author response: Yes, chart review was done for some variables. It is revised as follows: the questionnaire has three sections. The first section was about socio demographic characteristics of the participants. Secondly, patients, medical records were assessed to extract additional information concerning to the date of initiation of AEDs, types of epilepsy and related issues clinical issues. The third section consists the Quality Of Life in Epilepsy (QOLIE)-31questionner which is a survey question of health-related quality of life for adults (18 years or older) with epilepsy. Page 6-7.

Comment: The proportion of medication adherence is not included in the result section.

Author response: Included in table 2.

Abstract: Male sex, higher educational status, and age at onset of epilepsy having family history of epilepsy treatment follow-up duration, having uncontrolled seizure, pre-treatment number of seizures, poor adherence to anti-epileptic drugs, having moderate anxiety and depression, having sever anxiety and depression, believing that epilepsy is caused by evil spirit, drinking alcohol, and having other co morbidities were negative predictors; write these factors with OR & CI.

Author response: Revised as follows: Male sex (B = 4.34, 0.41, 8.27), higher educational status (B = 7.18, 1.39, 13.00) and age at onset of epilepsy (B = 0.237, 0.017, 0.45) were associated with increased health related quality of life score. On the other hand, family history of epilepsy (B = -4.78, -9.24,-0.33), uncontrolled seizure (B = -11.08, -15.11,-7.05), more than 5 pre-treatment number of seizures (B = -4.86, -8.91,-0.81), poor drug adherence (B = -11.65, -16.06,-7.23), having moderate (B = -4.526, -8.59, -0.46) to sever (B = -12.84, -18.30,-7.37) anxiety and depression, believing that epilepsy is caused by evil spirit (B = -7.04, -11.46,-2.61), drinking alcohol (B = -5.42, -10.72,-0.13), and having other co-morbidities (B = -9.35,-14.35,-4.36) were significantly negatively associated with the health related quality of life score among epileptic patients. Page 2.

Conclusion: Almost half of the epileptic patients have an overall weighted average health related quality of life score below the mean score level. This conclusion is opposite of your statement written on your result???? ‘’ More than half, 199 (50.4%) of epileptic patients have overall weighted average health related quality of life score above or equal to the mean score level’’.

Author response: Thank you, it is revised as: Only around half of the epileptic patients had good quality of life. In addition, multiple variables including family history, uncontrolled seizure, poor adherence, and having anxiety and depression were associated with health related quality of life among epileptic patients. Hence, targeting these variables in epilepsy management is recommended. Page 2.

Comment: The largest ethnic group in north Wollo is the Amhara (99.38%) and all other ethnic groups made up 0.62% of the population. Amharic is spoken as a first language by 99.28% of the population and the remaining 0.72% spoke other primary languages. 82.74% of the populations are Ethiopian Orthodox Christianity followers, and 17.08% are Muslims. I think this paragraph is un necessary for your study, instead add the number of epileptic or study populations in the study area and try to relate your study population with the study area!!!!

Author response: Revised as follows in Study setting, design, and period section. An institution-based cross-sectional was carried out in North Wollo zone public hospitals, Amhara region, Ethiopia. According to the 2007 Census conducted by the Central Statistical Agency of Ethiopia (CSA), North Wollo Zone has a total population of 1,500,303, of whom 752,895 are men and 747,408 women; with an area of 12,172.50 square kilometers (23). The study participants were recruited from Kobo primary hospital, Lalibela primary hospital, and Woldia Comprehensive Specialized hospital. The study was conducted from February, 01 to June 30, 2020. Methods section, first paragraph.

Comment: What is the deference between source population and study population on your study? If not you can use one of the two.

Author response: Revised: Source population: Adult patients (age ≥ 18 years), who have been diagnosed with epilepsy, and are on at least one anti-epileptic drug therapy in north Wollo zone public hospitals. Study population: Adult patients (age ≥ 18 years), who have been diagnosed with epilepsy and have been on regular follow- up for at least one year with at least one anti-epileptic drug therapy in selected public hospitals. Therefore, the epileptic patients who have been on regular follow in selected public hospitals were the study populations because; they are the actual population where samples were taken. Page, 5.

Comment: You have two objectives magnitude of quality of life and factors associated with it, but you didn’t show any thing about you calculates sample size for the second objective. How could you see this? Since it can be affect the validity of result.

Author response: Thank you for the comment. It is revised as follows: The sample size for this study was determined by using the single population proportion formula considering the assumptions: The proportion of epileptic patients being seizure free, 38.6%, taken from a study in Addis Ababa (p = 0.386) because it provides the maximum sample size among the factors for quality of life (25). Level of significance 5% (α = 0.05), Z α/2 =1.96 and margin of error 5% (d = 0.05). The sample size was calculated as follows:

no = Z (α/2)2*p (1-p)

 d2

Hence, the minimum required sample size was = (1.96)2*(0.386) (0.614)/ (0.05)2= 364. Adding 10% non-response rate, the final sample size appeared to be 401. However, in a paper to be published, we have thought that putting all the procedures of sample size calculation might not be important. Page, 5.

Comment: Finally which method of simple random sampling technique method do you used please try to clarify it!

Author response: Revised: Finally, study participants were recruited consecutively by assuming patient flow as random. Page, 5-6.

Comment: Result: Two hundred seventy four (60.4%). This information varies from the data on the table. 

Author response: It is corrected that more than half, (58.5%) of them were females. Page, 8.

Comment: Pre-treatment duration with epilepsy “≤ 12 month”: This is out of your inclusion criteria. See your inclusion criteria.

Author response: In our eligibility criteria: All adult epileptic patients (age ≥ 18 years), who have been on regular follow- up for at least one year with at least one anti-epileptic drug therapy were included.. However, the duration that patients live with epilepsy before seeking treatment is not an exclusion criterion. Before seeking treatment, patients might have lived with epilepsy for one year and above and this is not our inclusion criteria. What we excluded was patients who have less than one year treatments follow up period because, seizure freedom 1 year after AED treatment is suggested as a good predictor of long-term remission (reference 24). Page 4.

Comment: Your tool contains socio demographic characteristics of the participants. The second section covers a data extraction checklist on seizure control status and related issues. The third section consists the Quality Of Life in Epilepsy (QOLIE)-31questionner which is a survey question of health-related quality of life for adults (18 years or older) with epilepsy. Health related quality of life was measured based on total score of QOLIE-31questionner. A QOLIE- 31questionner consists 31 items categorized under seven domains covering the following concepts of health: Overall Quality of Life (2 items), Medication Effects (3 items), Energy Fatigue (4 items), Seizure Worry (5 items), Emotional Well-being (5 items), Cognitive Function (6 items), and Social Function (5 items). Therefor where is your table which is containing most parts of your study variables……..? I have got only socio-demographic and logistic regression table only. 

Author response: Thank you for your insightful comment. We have seen that only sociodemographic and some clinical variables were included in table one. Therefore, another table is included as table 2 to show the necessary Variables separately from table one. Page. 10. 

Comment: What is the difference between overall quality of life and overall weighted average quality of life? Which is better for your study? Try to make clear is for readers and use your best one! 

Author response: Each of the seven domains of QOL was scaled from 100%. Quality of life was one of the seven domains and it was measured by using 2 questions. And then, the overall weighted average quality of life score was calculated by summing the product of each of the seven domain scales times its weight and summing overall scales. Further details about the scoring of QOLIE-31 are available somewhere else (Reference 30; Quality of Life in Epilepsy inventory, QOLlE-31 (Version 1.0) Scoring Manual: Available at: https://www.rand.org/content/dam/rand/www/external/health/surveys_tools/qolie/qolie31_scoring.pdf, (Accessed on September 3, 2019).

Comment: “about 50.8% of the total variation in health related quality of life among epileptic patients was explained by the variables included in the model”. Is it appropriate model? Where is the other 49.2%. When we say it is a good model? 

Author response: R-squared that is the explained variation / Total variation always ranges between 0 and 100%. 0% indicates that the model explains none of the variability of the response data around its mean. On the other hand, 100% indicates that the model explains all the variability of the response data around its mean. In general, it is considered that the higher the R-squared, the better the model fits the data. However, it is not always true. In our study, the value of R-square is >50 percent and it is a good model as far as it explains more than 50% of the variations.

Comment: Linear regression analysis of factors for health related quality of life in epileptic patients. I didn’t get any information on the methodology part about linear regression model assumption were full filed or not!

Author response: Linear regression model assumptions, that that is variable measurement at the continuous level, the presence of linear relationship between the independent and the outcome variables, absence of significant outliers, independence of observations, homogeneity of variance, normally of distribution among the errors and absence of multicollinearity among the independent variables were assessed and all of them were fulfilled. Page, 7-8.

Comment: Author’s Contribution: AMK participates in all steps of the study from its commencement to writing. BBA, MWK, AD, AW, and MA have participated in reviewing the paper, analysis and interpretation of the data. All authors have read and approved the submission of the final manuscript. Include all this authors on the cover page! 

Author response: I apologize for failing to do this; all have been included in the first page.

---

## [Decision Letter · Decision Letter 1]

26 Jan 2021

PONE-D-20-34326R1

Quality of life and its associated factors among epileptic patients attending public hospitals in North Wollo Zone, Northeast Ethiopia: a cross-sectional study

PLOS ONE

Dear Dr. Kassie,

Thank you for submitting your manuscript to PLOS ONE. After careful consideration, we feel that it has merit but does not fully meet PLOS ONE’s publication criteria as it currently stands. Therefore, we invite you to submit a revised version of the manuscript that addresses the points raised during the review process.

Please try to improve as mush as possible data presentation as suggested.

furthermore, improve the language used.

We look forward to receiving your revised manuscript.

Kind regards,

Emilio Russo

Academic Editor

PLOS ONE

Reviewers' comments:

Reviewer's Responses to Questions

**Comments to the Author**

1. If the authors have adequately addressed your comments raised in a previous round of review and you feel that this manuscript is now acceptable for publication, you may indicate that here to bypass the “Comments to the Author” section, enter your conflict of interest statement in the “Confidential to Editor” section, and submit your "Accept" recommendation.

Reviewer #1: All comments have been addressed

Reviewer #3: (No Response)

2. Is the manuscript technically sound, and do the data support the conclusions?

Reviewer #1: Yes

Reviewer #3: Partly

3. Has the statistical analysis been performed appropriately and rigorously? 

Reviewer #1: Yes

Reviewer #3: Yes

4. Have the authors made all data underlying the findings in their manuscript fully available?

Reviewer #1: Yes

Reviewer #3: Yes

5. Is the manuscript presented in an intelligible fashion and written in standard English?

Reviewer #1: Yes

Reviewer #3: No

6. Review Comments to the Author

Reviewer #1: This paper has a potential to be accepted, the authors have adequately addressed my comments raised in a previous round of review and I feel that this manuscript is now acceptable for publication since the manuscript technically sound, and do the data support the conclusions

Reviewer #3: Authors deeply improved the manuscript in the revised version, addressing previous comments. However, further modifications are needed before publication:

1 - Abstract: please report data in results as: (B= ..., 95%CI ...; p ...)

2 - Epilepsy nomenclature should be allineate with new ILAE definition (2018), for instance: partial seizures should be referred as focal seizures (in the text as well in the tables), etc.

3 - Background, pag 3, lines 1-6: please provide a more detailed/technical characterization of epilepsy.

4 - Please change Antiepileptic drugs (AEDs) in "antiseizure medications (ASMs) throughout the manuscript.

5 - "with an area of 12,172.50 square kilometers" not relevant information.

6 - Please combine study setting, Source population and study population paragraphs, avoiding redundant data.

7 -Please combine and synthetize "data collection technique and tools" and "Data quality assurance" paragraphs for better clearness and readability.

8 - "Ethical approval" - the local Ethical Committee has provided an approvation number/protocol? If so, please state it. Please remove sentences as: "All participants were asked for their 12 willingness to participate in the study and were told that it will not have any risk on them", these information are implicit in the informed consent.

9 - Tables 1and 2 - please report frequency and percentage in the same column as: frequency (percentage), for instance sex male 231 (58.5). For dichotomic data (e.g. sex, alcohol drinking habit, smoke, chew chat) please report in the table one of the variable, eg. smoke 23 (5.8). remove blankets for age data.

10- table 1 and 2 - please remove "2020" and "North Wollo Zone public hospitals 2020" from the titles.

11 - Please provide Nagelkerke R2 for multivariate regression analysis.

12 - recheck the manuscript for typos.

7. PLOS authors have the option to publish the peer review history of their article (what does this mean?). If published, this will include your full peer review and any attached files.

Reviewer #1: **Yes: **Agumas Fentahun Ayalew

Reviewer #3: No

---

## [Author Response · Author response to Decision Letter 1]

3 Feb 2021

To: Plose One Editorial Team

Subject: Submitting a revised research manuscript for publication 

Dear Sir /Madam 

Greetings,

 I am Ayelign Mengesha from Woldia University, Ethiopia. I hereby submit a revised manuscript entitled “Quality of life and its associated factors among epileptic patients attending public hospitals in North Wollo Zone, Northeast Ethiopia: a cross-sectional study” to be published at Plose One. 

 First of all I would like to express my deepest gratitude on behalf of the authors for the very detail, genuine and constructive comments raised by the reviewer/s and, or the editor which I believe has improved the manuscript tremendously. We believe that we have addressed the points raised in the review process properly. The modifications and, or corrections made during the revision are clarified point by point below. 

 Yours sincerely,

 On behalf of the authors

General corrections/modifications:

Grammatical, spelling and punctuation mark issues are corrected with grammerly proof reading tool and with intense revision by the authors.

The other corrections/modifications are described in detail under point by point responses below.

General comments: Authors deeply improved the manuscript in the revised version, addressing previous comments. However, further modifications are needed before publication:

Author response: I thank you very much on behalf of the co-authors for your genuine and constructive comments. We have revised the manuscript intensively based on the comments provided. I particularly would like to thank you for the updated information on the nomenclature’s of epilepsy/seizure disorders.

Comment1: - Abstract: please report data in results as: (B= ..., 95%CI ...; p ...)

Author response: Incorporated as suggested.

Comment 2 - Epilepsy nomenclature should be allineate with new ILAE definition (2018), for instance: partial seizures should be referred as focal seizures (in the text as well in the tables), etc.

Author response: Revised accordingly. 

Comment 3 - Background, pag 3, lines 1-6: please provide a more detailed/technical characterization of epilepsy.

Author response: 

It is revised as follows. Epilepsy is a chronic and non-communicable disease of the brain. It is characterized by recurrent seizure episodes involving part of the body or the entire body. Seizure episodes are due to the release of excessive electrical discharges from a group of brain cells (1). The seizures can present in different forms ranging from a brief lapse of attention or muscle jerks to severe and prolonged convulsions. The episode also varies in frequency, ranging from less than one per year to several seizures per day. Globally, more than 50 million people are estimated to have active epilepsy and the annual cumulative incidence rate estimated at 67.77 per 100,000 persons.

Comment 4 - Please change Antiepileptic drugs (AEDs) in "antiseizure medications (ASMs) throughout the manuscript.

Author response: Replaced/changed throughout the manuscript.

Comment 5 - "with an area of 12,172.50 square kilometers" not relevant information.

Author response: The sample was taken based on the number and/or patient flow in the study area. So, it is fair to say that the sample can be representative of the patients who were attending the hospitals at the time of the study. But, it does not mean it is representative of the population of north Wollo zone.

Comment 6 - Please combine study setting, Source population and study population paragraphs, avoiding redundant data.

Author response: They are combined. 

Comment 7 -Please combine and synthetize "data collection technique and tools" and "Data quality assurance" paragraphs for better clearness and readability.

Author response: They are combined. 

Comment 8 - "Ethical approval" - the local Ethical Committee has provided an approbation number/protocol? If so, please state it. Please remove sentences as: "All participants were asked for their 12 willingness to participate in the study and were told that it will not have any risk on them", these information are implicit in the informed consent.

 It is re-written is follows: Ethical clearance was obtained from Woldia University institutional review board with an approval number of wldu/012/2020 and support letters were written to the selected public hospitals and concerned others to obtain permission and cooperation during the data collection process. Written informed consent was obtained from each study participant before the data collection process. Confidentiality of the information was preserved and the privacy of the respondents was maintained by making the questionnaires anonymous and putting them in secured places after the data were collected. 

Author response: The section "All participants were asked for their 12 willingness to participate in the study and were told that it will not have any risk on them" is removed.

Comment 9 - Tables 1and 2 - please report frequency and percentage in the same column as: frequency (percentage), for instance sex male 231 (58.5). For dichotomic data (e.g. sex, alcohol drinking habit, smoke, chew chat) please report in the table one of the variable, eg. smoke 23 (5.8). remove blankets for age data.

Author response: Revised as suggested.

Comment 10- table 1 and 2 - please remove "2020" and "North Wollo Zone public hospitals 2020" from the titles.

Author response: removed 

Comment 11 - Please provide Nagelkerke R2 for multivariate regression analysis.

Author response: Revised as follows: under table 3, Model fitness: (R2) = 50.8% and in text above table 3.

Comment 12 - recheck the manuscript for typos. 

Author response: Revised with co-authors and grammerly proof editing tool. Thank you very much.

---

## [Editor Report · Decision Letter 2]

5 Feb 2021

Quality of life and its associated factors among epileptic patients attending public hospitals in North Wollo Zone, Northeast Ethiopia: a cross-sectional study

PONE-D-20-34326R2

Dear Dr. Kassie,

We’re pleased to inform you that your manuscript has been judged scientifically suitable for publication and will be formally accepted for publication once it meets all outstanding technical requirements.

Kind regards,

Emilio Russo

Academic Editor

PLOS ONE
---

## [Editor Report · Acceptance letter]

12 Feb 2021

PONE-D-20-34326R2 

Quality of life and its associated factors among epileptic patients attending public hospitals in North Wollo Zone, Northeast Ethiopia: a cross-sectional study 

Dear Dr. Kassie:

I'm pleased to inform you that your manuscript has been deemed suitable for publication in PLOS ONE. Congratulations! Your manuscript is now with our production department. 

Kind regards, 

on behalf of

Prof Emilio Russo 

Academic Editor

PLOS ONE